



# Retrieved wind speed from the Orbiting Carbon Observatory-2

Robert R. Nelson[1], Annmarie Eldering[1], David Crisp[1], Aronne J. Merrelli[2], and Christopher W. O'Dell[3]

[1]Jet Propulsion Laboratory, California Institute of Technology, Pasadena, CA, USA
[2]Space Science and Engineering Center, University of Wisconsin-Madison, Madison, WI, USA
[3]Cooperative Institute for Research in the Atmosphere, Fort Collins, Colorado, USA

**Correspondence:** Robert R. Nelson (Robert.R.Nelson@jpl.nasa.gov)

**Abstract.**

Satellite measurements of surface wind speed over the ocean inform a wide variety of scientific pursuits. While both active and passive microwave sensors are traditionally used to detect surface wind speed over water surfaces, measurements of reflected sunlight in the near-infrared made by the Orbiting Carbon Observatory-2 (OCO-2) are also sensitive to the wind speed. In this work, retrieved wind speeds from OCO-2 glint measurements are validated against the Advanced Microwave Scanning Radiometer-2 (AMSR2). Both sensors are in the international Afternoon Constellation, allowing for a large number of co-located observations. Several different OCO-2 retrieval algorithm modifications are tested, with the most successful being a single-band Cox-Munk-only model. Using this, we find excellent agreement between the two sensors, with OCO-2 having a small mean low bias against AMSR2 of -0.22 m/s, an RMSD of 0.75 m/s, and a correlation coefficient of 0.94. Although OCO-2 is restricted to clear-sky measurements, potential benefits of its higher spatial resolution relative to microwave instruments include the study of coastal wind processes, which may be able to inform certain economic sectors.

Copyright statement

## 1 Introduction

Surface wind speed has been measured by satellites going back nearly half a century. These measurements have proven extremely valuable in improving weather and climate models while advancing our understanding of oceanic and atmospheric physics. Both active and passive sensors are used to estimate wind speeds and these measurements are typically made in the microwave in order to penetrate through clouds and most precipitation. Technically, these satellites are sensitive to the surface roughness. Ocean surfaces respond quickly to the movement of the air above them, and thus the surface roughness pattern is a function of both wind speed and wind direction. This wind speed measurement technique is limited in that it does not work over land or ice surfaces.

Active instruments, including scatterometers (e.g. SeaWinds (Spencer et al., 2000), ASCAT (Figa-Saldaña et al., 2002), RapidScat (Durden and Perkovic-Martin, 2017)), altimeters (e.g. SEASAT (Born et al., 1979), SARAL-AltiKa (Lillibridge et al., 2014), and synthetic aperture radars (e.g. RADARSAT-1 (Parashar et al., 1993), ALOS PALSAR (Rosenqvist et al., 2007) estimate wind speed and sometimes direction by sending electromagnetic pulses to the surface then detecting and characterizing





the backscattered radiation. Wind speed, but not direction, can also be estimated from measurements of radiation obtained by passive microwave instruments that operate at a variety of frequencies. The characteristics of this radiation depend on wind-induced effects on surface roughness and the production of white caps (Bourassa et al., 2010) so typically a radiative transfer model is used to estimate wind speed from these emission characteristics (e.g. Wentz, 1997; Meissner and Wentz, 2012.

Examples of passive sensors include the Special Sensor Microwave Imager (SSM/I; Hollinger et al., 1990; Wentz, 1997), the Special Sensor Microwave Imager Sounder (SSMIS), the Tropical Rainfall Mission Microwave Imager (TMI; Wentz, 2015), the Global Precipitation Mission (GMI; Draper et al., 2015; Wentz and Draper, 2016), and the Advanced Microwave Scanning Radiometers (AMSR-E and AMSR2; Imaoka et al., 2010). They all measure at microwave frequencies from 6 GHz to 37 GHz at both vertical and horizontal polarizations, allowing for the removal of atmospheric attenuation effects. The spatial resolution

of these passive sensors typically ranges from 20-35 km.

    In addition to missions specifically designed to measure wind speed, many space-borne sensors that measure reflected sunlight in the visible or near-infrared must have some way of accounting for reflection off of specular surfaces such as the ocean. The Orbiting Carbon Observatory-2 (OCO-2, (Crisp et al., 2008)) is one such instrument. It measures reflected sunlight in three near-infrared bands and uses a Cox-Munk (Cox and Munk, 1954) sea surface slope model to estimate reflectance when

over water surfaces. These reflectances are primarily a function of illumination, viewing geometry, and wind speed. However, no effort has been made to validate the wind speed estimates from OCO-2 until now.

    Section 2 discusses the two primary datasets used in this validation study, OCO-2 and AMSR2. Section 3 describes the OCO-2 retrieval and, specifically, the wind speed derivation. Section 4 presents results from four different OCO-2 retrieval variants and show how they compare to AMSR2 wind speeds. Finally, Sec. 5 summarizes the results and discusses the potential

scientific utility of OCO-2 wind speed measurements.

## 2   Data

In this work we compare wind speed estimates from spectroscopic observations from OCO-2 to passive microwave observations from the Advanced Microwave Scanning Radiometer 2 (AMSR2) on the Japanese Global Change Observation Mission for Water-1 (GCOM-W1) satellite. Both satellites are in a unique sun-synchronous polar orbit know as the Afternoon Constellation

(L'Ecuyer and Jiang, 2010) which enables excellent co-location in both time and space. AMSR2 is approximately 5 minutes behind OCO-2 and has a swath width of 1450 km resulting in near global coverage every day. OCO-2 measures eight adjacent footprints, each with a resolution of approximately 1.25 km by 2 km at nadir, resulting in a swath width of about 10 km. It has a repeat cycle of 16 days and makes about one million observations a day. Over water surfaces, which are relatively dark in the near-infrared, OCO-2 changes its viewing geometry in order to view a surface track near the much brighter sun glint spot

(rather than nadir) in order to significantly increase the measured signal.

    The AMSR2 wind speed product used for validation in this work is from Remote Sensing Systems (RSS; Meissner and Wentz, 2012). RSS provides two standard rain-free radiometer wind speed products: low frequency and medium frequency. Both are available on a 0.25 degree latitude-longitude grid. The low frequency product uses microwave channels at 10.7,





18.7, 23.8, and 36.5 GHz while the medium frequency product uses only 18.7, 23.8, and 36.5 GHz. Each product has its own benefits and drawbacks. For example, the low frequency product is less impacted by the atmosphere and rain, but is affected by radio frequency interference in the 10.7 GHz channel as well as sun glint effects. The medium frequency product has a higher effective spatial resolution and is less affected by ice and land contamination, but is slightly noisier than the low

frequency product. Because of this, the comparisons presented here use the medium frequency AMSR2 wind speed as the primary reference product. However, we briefly discuss results from the low frequency product in Sec. 4.4.

No temporal threshold was needed for co-locating OCO-2 and AMSR2, as the nature of both satellites' scanning patterns results in the difference in time between a given OCO-2 footprint and an AMSR2 grid cell ranging from 6 minutes behind to 4 minutes ahead. While each OCO-2 footprint typically falls within a 0.25° by 0.25° AMSR2 grid cell, a distance threshold of <

0.1 degrees was implemented to ensure that both instruments were observing approximately the same location. As GCOM-W1 was launched in 2012 and OCO-2 in 2014, the co-located data used in this study ranged from September 2014 to January 2019.

The accuracy of the AMSR2 wind speed product is fairly well characterized and is on the order of 1-1.5 m/s for wind speeds of 0-15 m/s (Wentz, 1997; Mears et al., 2001; Kachi et al., 2013; Ebuchi, 2014; Ricciardulli and Wentz, 2015; Wentz et al., 2017). RSS inter-calibrates several microwave radiometers and thus conclusions about errors for one satellite are typically true

for the entire suite of radiometers in a given study. Other validation work includes Kachi et al. (2013), who compared them to buoy wind speeds and found a root mean square deviation (RMSD) of 1.12 m/s. Additionally, Ebuchi (2014) estimated the RMSD against buoy data to be 0.99 m/s for the RSS low frequency product and 1.06 m/s for the medium frequency product. In general, the accuracy of microwave radiometers tends to degrade when viewing rainy scenes. However, OCO-2 only returns useful data in cloud-free conditions so this should not be an issue for this comparison because the co-location in space should

be close enough such that both instruments are viewing cloud- and rain-free scenes. Finally, while we recognize that buoys are generally considered the best validation metric, we forego them here in favor of AMSR2 because of its excellent co-location with OCO-2 in both time and space.

## 3   OCO-2 Retrieval Algorithm

OCO-2 measures reflected sunlight in three near-infrared bands: the molecular oxygen ($O_2$) A-band at 0.765 $\mu$m, a weakly

absorbing carbon dioxide ($CO_2$) band at 1.61 $\mu$m, and a strongly absorbing $CO_2$ band at approximately 2.06 $\mu$m. Coincident, high resolution ($\lambda/\Delta\lambda \sim$ 17,000-19,000) spectra collected in these three channels are combined to form soundings that are analyzed with a remote sensing retrieval algorithm to estimate the the column-averaged dry-air mole fraction of $CO_2$ ($X_{CO_2}$) along with several other atmosphere and surface state properties that affect the measured radiances. In short, the retrieval algorithm starts with an assumed state vector containing a priori values and corresponding uncertainties and uses a full-physics

surface-atmosphere radiative transfer model and an instrument model to simulate the observed spectra. It then uses optimal estimation (Rodgers, 2000) to update the state vector properties to minimize a cost function that reduces differences between measured and modeled radiance spectra within the constraints of the specified uncertainties. The final result is an optimized state vector, which is a weighted combination of information from the measurements themselves and the a priori values, and



an a posteriori uncertainty for each state vector element. Full details of the process and the state vector elements can be found in O'Dell et al. (2018). The current OCO-2 algorithm, version 9 (B9) of the Atmospheric Carbon Observations from Space (ACOS; O'Dell et al., 2012; Crisp et al., 2012), contains a state vector with approximately 55 elements including a 20-level $CO_2$ profile, band-dependent albedos and albedo slopes, surface pressure, five cloud and aerosol types, etc. Over ocean, a wind

speed scalar is also retrieved.

## 3.1 Wind Speed Retrievals

Over liquid water surfaces, the OCO-2 retrieval algorithm assumes that the surface reflectance could be simulated by a combination of two surface types: a Cox-Munk distribution of planar facets and a Lambertian surface. The Cox-Munk parameterization (Cox and Munk, 1954) was developed from the brightness distribution in 29 aerial photographs of sunlight reflected off of

the ocean near Hawaii over a 20-day period. Their observations gave a distribution of wind-generated sea surface slopes that could be approximated by a Gaussian and expressed by a Gram-Charlier expansion. They found that the mean square slope parameter, which describes the surface roughness in their photographs, could be related to wind speed using a function of the form:

$$\sigma_{cm}^2 = 0.003 + 5.12 \times 10^{-3} U \tag{1}$$

where $U$ is wind speed in m/s and $\sigma_{cm}^2$ is the mean square slope. This empirical model describes the probability that the sea surface will be oriented to cause sun glint, depending on the wind speed. Further details can be found in Cox and Munk (1954), Su et al. (2002), Kay et al. (2009), Monzon et al. (2006), and others. The Cox-Munk parameterization requires a refractive index in each band in order to produce an appropriate reflectance. Values for water are used: 1.331, 1.318, and 1.303 in the $O_2$ A-Band, weak $CO_2$, and strong $CO_2$ bands, respectively (Hale and Querry, 1973) with an adjustment made for sea water

(Friedman, 1969; McLellan, 1965; Sverdrup et al., 1942). The Cox-Munk model was developed from measurements made at 41 ft (12.5 m), while surface wind speed products, including those provided by RSS, are typically reported at 10.0 m. Thus, we use a log wind profile assumption to convert the 12.5 m wind speed values to 10.0 m above the surface:

$$U = \frac{u_*}{k} ln\left(\frac{z}{z_0}\right) \tag{2}$$

where $u_*$ is the friction velocity, $z_0$ is the aerodynamic roughness length, and $k$ is the Von Karman constant. Rearranging

and solving for the wind speed at 10.0 m with a $z_0$ of 0.009 (Stull, 1988) gives a scaling factor of 0.9766 to convert winds at 12.5 m to 10.0 m.

While the Cox-Munk surface parameterization should be sufficient to describe reflection off of a water surface, a Lambertian component was added because the ACOS retrieval is unable to perfectly fit the continua in all three bands with one free parameter (wind speed) (Crisp et al., 2017). Thus, a Lambertian albedo and albedo slope (a linear slope across the band added

to the albedo magnitude) are solved for in all three bands. This results in seven variables being used over water to describe the





surface. Because the Lambertian component is only being included to make a small difference in the fit (the Cox-Munk wind speed should do most of the work), the a priori albedo values are set to 0.02. Additionally, the strong $CO_2$ band albedo is fixed because various tests revealed that the retrieval often wanted to solve for negative albedos in that band. The likely reason for this is a 6-8% overestimate in the solar flux, which will be fixed in the upcoming version of ACOS. The current solution to

the issue is to simply not let it retrieve that value. The 1-sigma a priori uncertainty on the albedos in the $O_2$ A-band and weak $CO_2$ bands is 0.2. The albedo slopes in all three bands have a prior value of 0.0 and a prior uncertainty of 1.0. The a priori wind speed is taken from the Goddard Earth Observing System Model, Version 5 Forward Processing for Instrument Teams (GEOS-5 FP-IT; Rienecker et al., 2008) with a 1-sigma a priori uncertainty of 6.325 m/s.

## 4 Results

We evaluated the wind speed performance of the production OCO-2 ACOS B9 retrieval algorithm along with three modifications to this algorithm. All of the OCO-2 wind speed measurements were derived from sun glint measurements over water and have been scaled from 12.5 m to 10.0 m, as discussed in Sec. 3.1. Additionally, the B9 lite file $X_{CO_2}$ quality flag was applied in order to remove poor quality soundings. Details can be found in (O'Dell et al., 2018) and (Taylor et al., 2016), but in general the filtering process is designed to remove cloudy and aerosol-laden scenes and scenes with low signal levels. The

three additional tests were an update of the solar continuum (Sec. 4.2), a three-band Cox-Munk-only retrieval (Sec. 4.3), and a single-band Cox-Munk-only retrieval (Sec. 4.4).

### 4.1 B9 Wind Speed

The OCO-2 B9 wind speed comparison to the AMSR2 wind speed is shown in Fig. 1. The benefit of the two instruments being adjacent in time and space can be seen, as over 44 million co-located measurements are plotted. There is generally low scatter

and good agreement, but the OCO-2 estimates have a high bias that increases at higher wind speeds. The RMSD is 2.56 m/s with a mean positive bias of 1.8 m/s. Measurements below 1.5 m/s for OCO-2 were filtered out in this B9 dataset because they were correlated with poor quality $X_{CO_2}$ retrievals. However, the difference between retrieved OCO-2 and AMSR2 wind speeds is not correlated with $X_{CO_2}$ errors (not shown). This was also checked for the upcoming ACOS B10 retrieval, with no correlation being found.

### 25 4.2 TSIS-SIM Solar Fluxes

For all OCO-2 ACOS product versions before B10, the top-of-atmosphere solar flux spectrum was derived by convolving a high-resolution empirical solar line transmission spectrum (Toon, 2016) with a radiometrically-calibrated solar continuum fit to the ATmospheric Laboratory for Applications and Science-3 (ATLAS-3) SOLar SPECtrum (SOLSPEC), which flew on the Space Shuttle (Thuillier et al., 2003). More recent measurements of the solar spectra from instruments deployed on the

International Space Station (ISS) (Meftah et al., 2017) show that the ATLAS-3 SOLSPEC results overestimate the fluxes by 4-8% in the OCO-2 $CO_2$ bands.





As a part of the ACOS B10 development, new solar flux spectra were generated to better estimate of the top-of-atmosphere solar flux. This update replaces the ATLAS-3 SOLSPEC continuum with a fit to the new reference solar spectrum based on data from the Total and Spectral Solar Irradiance Sensor (TSIS) Spectral Irradiance Monitor (SIM) onboard the ISS. The OCO-2 continuum values were then scaled to match the new TSIS data, with the hope that a better band-to-band solar calibration

would lead to a reduction in the Lambertian component over water surfaces and general improvements to the retrieval otherwise. Figure 2 demonstrates how the TSIS solar continuum affects the retrieved wind speed over water surfaces. Compared to Fig. 1, we see a reduction in the overall wind speed bias from +1.81 m/s to +1.15 m/s and a small improvement in the RMSD and scatter. The TSIS solar continuum test was run on a relatively small dataset, but the statistics are similar when comparing the difference between OCO-2 B9 and AMSR2 on a comparably sampled dataset.

Despite the improvement in scatter and bias, the bias pattern in retrieved wind speed against AMSR2 persists. Figure 3 demonstrates that this high bias is strongly correlated with low signal levels. The $O_2$ A-Band is plotted here, but the same relationship exists for the weak and strong $CO_2$ bands.

## 4.3  Three-Band Cox-Munk-Only

The next experiment was designed to test the hypothesis that solving for both the wind speed and a Lambertian component

was inducing the mean high bias relative to AMSR2. This is because in the Cox-Munk plus Lambertian setup the retrieval has multiple ways to adjust the radiances to match the measured spectra. Specifically, it can adjust the wind speed, but also adjust the Lambertian albedos and albedo slopes. Figure 3 suggests that the Cox-Munk plus Lambertian component results in the retrieval of erroneously large wind speed when the signals are low (i.e. when the wind speed is high). Here, we turn off the Lambertian component and force the retrieval to solve for one wind speed to fit the continuum for all three bands over water

surfaces. Figure 4 shows that there is now a low bias of approximately 1 m/s but that 89.8% of the retrievals fail to converge.

Of note, Fig. 5 demonstrates that removing the Lambertian component in the surface reflectance parameterization greatly reduces the dependency on signal seen in Fig. 3. The noisier data in Fig. 5 relative to Fig. 3 is mostly due to a significant reduction in converged retrievals for this three-band Cox-Munk-only test.

## 4.4  Single-Band Cox-Munk-Only

The final test was designed to build upon the previous results and examine whether a single-band Cox-Munk-only retrieval could perform better compared to AMSR2, with the idea that one retrieved wind speed should be sufficient to fit the continuum of one OCO-2 band. In this test, only the $O_2$ A-band was used. The $CO_2$ retrieval was disabled, as neither of the $CO_2$ bands were used. Empirical orthogonal functions, which are part of the usual ACOS state vector, were also disabled to create as simplistic of a retrieval as possible. Figure 6 shows the results. The number of successful retrievals is much improved relative to the

three-band version (Sec. 4.3), with 91.0% meeting the convergence criteria. In addition to using the B9 quality flag, additional filtering was employed to remove a small number of highly erroneous retrievals. The difference between the retrieved surface pressure and the prior surface pressure was filtered on to exclude values outside of $\pm$ 8 hPa, retrieved ice cloud heights greater





than 0.14 were removed, and airmass factors greater than 2.8 were removed. The bias against AMSR2 is reduced to -0.22 m/s with an RMSD of 0.75 m/s and correlation coefficient of 0.94.

This same single-band Cox-Munk-only test was repeated using the weak $CO_2$ and strong $CO_2$ channels independently, with mixed results. For the weak $CO_2$ and strong $CO_2$ versions, respectively, the biases against AMSR2 were -0.90 m/s and -0.74 m/s and the RMSDs were 1.27 m/s and 1.27 m/s. Finally, comparison statistics were regenerated but using the RSS low frequency product (as discussed in Sec. 2). The statistics and shape of the distribution are similar, with a slightly worse bias (-0.31 versus -0.22 m/s) but somewhat improved scatter (0.67 versus 0.72 m/s) and linear fit slope (0.90 versus 0.94).

## 5 Discussion and Conclusions

Here, we assessed how well near-infrared observations of reflected sunlight from OCO-2 could be used to estimate surface wind speeds in cloud-free regions. Table 1 gives a statistical summary of the retrievals tested.

It was found that the operational product (ACOS B9) is high biased against AMSR2, with the bias increasing at higher wind speeds. The inclusion of an updated solar continuum from the TSIS instrument onboard the ISS improved the comparison slightly, but the high bias remained. The removal of the Lambertian component of the state vector resulted in the majority of retrievals failing to converge. This is probably because one wind speed is insufficient to fit the continuum radiances of all three OCO-2 bands, each with their own small calibration errors. A single-band Cox-Munk-only retrieval using the $O_2$ A-band with the updated solar continuum and a small height adjustment gives wind speeds that compare very well to AMSR2, with an RMSD of 0.75 m/s and a correlation coefficient of 0.94. Importantly, the retrieved wind speed shows better agreement to AMSR2 than the GEOS-5 FP-IT wind speed used as the OCO-2 meteorological prior which has an RMSD of 1.18 m/s compared to AMSR2. The weak $CO_2$ and strong $CO_2$ versions of the single-band Cox-Munk-only test resulted in worse scatter and bias compared to the $O_2$ A-Band version. This result may be explained by small uncorrected calibration errors in those bands. Additionally, while the two RSS wind speed products (medium and low frequency) give slightly different statistics when comparing the instruments, the conclusion that a modified OCO-2 retrieval can accurately and precisely measure wind speed holds true.

Another possible contribution to the differences between OCO-2 and AMSR2 is that the OCO-2 glint off-pointing strategy is not optimized to be maximally sensitive to the wind speed. This is because OCO-2 off-points further away from the glint spot at higher viewing angles while simultaneously the actual glint spot gets larger in size with faster wind speeds. This results in situations where windier scenes can be brighter than calm scenes at certain OCO-2 viewing angles. Ideally, OCO-2 would point directly at the glint spot to avoid this issue, but this risks damaging the instrument.

To extend the analysis beyond global statistics, Fig. 7 shows the difference between OCO-2 and AMSR2 wind speeds as a function of time and latitude. It shows a latitude dependance of the differences. There are multiple hypotheses that could explain this pattern. The Cox-Munk parameterization was developed on measurements restricted to solar zenith angles less than 35 degrees, while OCO-2 views the glint spot at solar zenith angles from around 16 degrees to upwards of 70 degrees and looks further away from the glint spot as the angle increases. As the solar zenith angle is closely tied to latitude, this could



explain the low bias at high latitudes, and indeed some of the low biased retrievals are removed with the airmass filter described in Sec. 4.4. Besides viewing geometry, numerous studies have suggested that the relationship between reflectivity and wind speed derived by Cox and Munk (1954) depends on atmospheric stability (Haimbach and Wu, 1985; Hwang and Shemdin, 1988; Shaw and Churnside, 1997). They found that stable air suppressed ripples and subsequently would produce a lower wind

speed estimate, and vice versa. However, additional study is needed to determine if, for example, the OCO-2 high bias seen in parts of the tropics in Fig. 7 is associated with unstable air.

There have been many other critiques and attempts to improve the Cox-Munk parameterization (e.g. Wu, 1972, 1990; Wentz, 1976; Ebuchi and Kizu, 2002; Tatarskii, 2003; Bréon and Henriot, 2006. See Zhang and Wang (2010) for an overview.) but in general it is still widely used in remote sensing to describe the reflection of sunlight off of water. Improvements to the original

parameterization, including new ways of fitting the data and the inclusion of additional measurements, could explain some of the remaining differences between OCO-2 and AMSR2, but it is beyond the scope of this work to implement them.

There are a number of potential applications for accurate and precise wind speeds from OCO-2. The small footprint size allows for the detection of wind speed closer to coasts than microwave radiometers, which currently have resolutions on the order of 25 km and are thus unable to view close to coastlines. Bourassa et al. (2019) note that "hydraulic expansion fans in

the marine boundary layer near capes and points (Winant et al., 1988; Rahn and Garreaud, 2014; Parish et al., 2016), coastally-trapped wind reversals (Nuss et al., 2000), and along-shore wind jets confined by coastal mountains can have cross-coast scales of 5-10 km or smaller" and that we have limited knowledge of all of these features. High resolution wind speed measurements would be able to detect winds much closer to coastlines and advance our understanding of these processes. Several economic sectors could also benefit from near-coast wind speed measurements, including oil-spill response, wind energy forecasting, and

search and rescue operations. Bourassa et al. (2019) write that the current plan to enhance spatial coverage from microwave sensors is to reduce onboard data averaging, but a wide-swath OCO-2-like instrument could provide highly accurate wind speed measurements near the coast in clear-sky conditions, depending on the viewing capabilities. Further study is needed to confirm the quality of these OCO-2 near-coast measurements, for example by comparing to buoy wind speeds, as shallow waters and turbidity may impact the retrieval in the $O_2$ A-band (the $CO_2$ bands will be less impacted, as they have penetration depths of

less than 1 mm.) It should be noted that certain active wind speed sensors, specifically altimeters, have footprints on the order of 1-10 km (Zieger et al., 2009). However, their coverage is limited and thus OCO-2 would provide useful complementary measurements.

Additionally, wind speed measurements at different times of day could help constrain the diurnal cycle of ocean winds. OCO-3, which is the backup of OCO-2 and currently deployed on the ISS, also makes glint measurements but these wind

speed measurements span the entire daytime due to the ISS's precessing orbit. Additional work is needed to validate the retrieved wind speed from OCO-3, but the instrument has characteristics very similar to OCO-2 and thus it is likely that the conclusions found here are also valid for OCO-3. Finally, this work will inform a number of future OCO-2-like instruments, such as MicroCarb (Buil et al., 2011) and the ambitious Copernicus $CO_2$ Monitoring Mission (Sierk et al., 2019).



*Code and data availability.* The OCO-2 L2 Full Physics Code is open source and available on GitHub https://github.com/nasa/RtRetrievalFramework (last access: 7 May 2020), and corresponding User's Guide is available at http://nasa.github.io/RtRetrievalFrameworkDoc/ (last access: 7 May 2020). All of the OCO-2 data products are publicly available through the NASA Goddard Earth Science Data and Information Services Center (GES DISC) for distribution and archiving (http://disc.sci.gsfc.nasa.gov/OCO-2; last access: 7 May 2020).

5  *Author contributions.* RN designed and performed the experiments. AE helped supervise the project. DC, AM, and CO contributed to the interpretation of the results. RN wrote the manuscript with contributions from all authors.

*Competing interests.* The authors declare that they have no conflict of interest.

*Acknowledgements.* The research was carried out at the Jet Propulsion Laboratory, California Institute of Technology, under a contract with the National Aeronautics and Space Administration (80NM0018D0004).





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





**Table 1.** Summary of the retrieval tests performed. Percent converged represents the fraction of converged soundings for the three retrieval variants, which were run on a set of soundings that were chosen to contain only high quality soundings. The TSIS solar test was run on a slightly different set of soundings than the three- and single-band Cox-Munk-only tests. The ACOS B9 retrieval was run on a very large set of less filtered soundings and thus the percent of converged soundings is not comparable to the other three tests.

| Retrieval | Percent Converged | Bias [m/s] | $\sigma$ [m/s] | RMSD [m/s] |
|---|---|---|---|---|
| ACOS B9 | N/A | 1.807 | 1.818 | 2.563 |
| TSIS Solar | 93.6 | 1.149 | 1.685 | 2.039 |
| Three-Band Cox-Munk-Only | 10.2 | -0.974 | 0.943 | 1.356 |
| SIngle-Band Cox-Munk-Only | 91.0 | -0.217 | 0.721 | 0.753 |

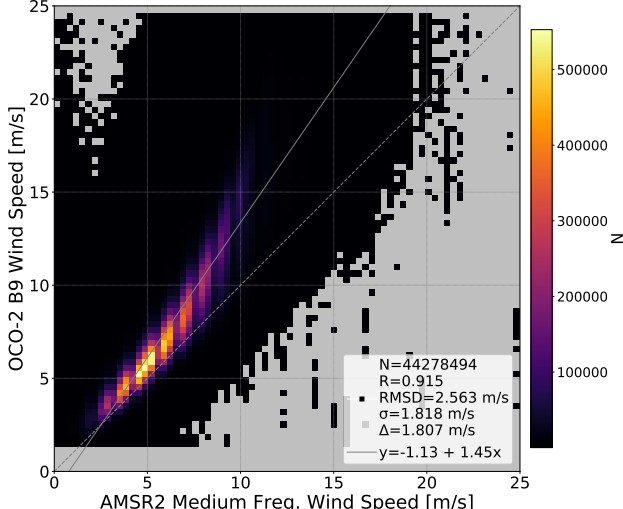

**Figure 1.** Heatmap of OCO-2 B9 wind speed compared to AMSR2 medium frequency wind speed. The B9 lite file quality flag has been applied, along with a height scaling.





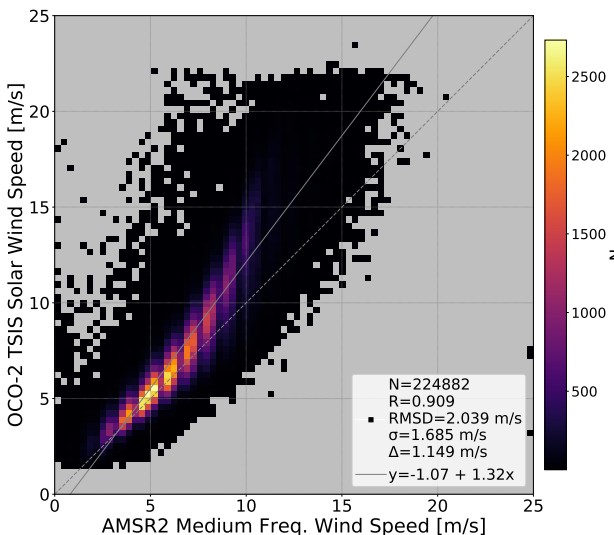

**Figure 2.** Heatmap of OCO-2 TSIS solar wind speed compared to AMSR2 medium frequency wind speed. The B9 lite file quality and preliminary B10 quality flags have been applied, along with a height scaling.

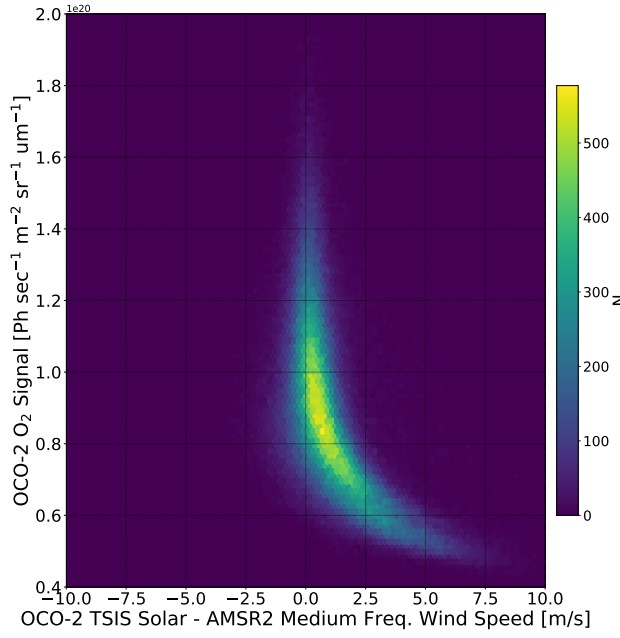

**Figure 3.** Heatmap of OCO-2 $O_2$ A-band signal compared to wind speed difference (OCO-2 TSIS solar - AMSR2 medium freq.)





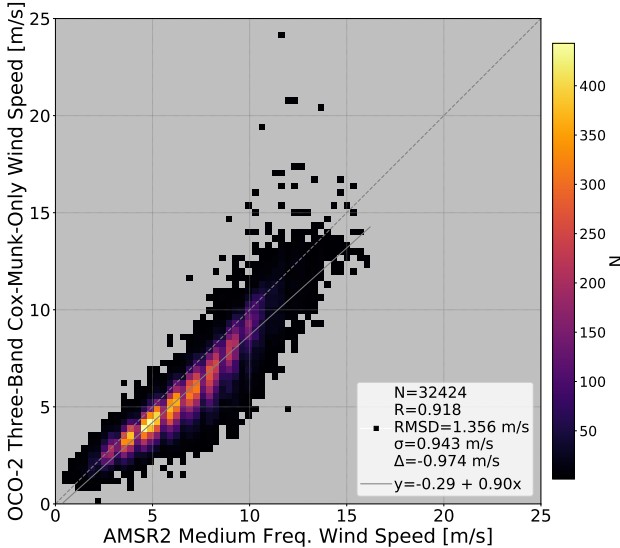

**Figure 4.** Heatmap of OCO-2 three-band Cox-Munk-only wind speed compared to AMSR2 medium frequency wind speed. The B9 lite file quality flag and height scaling have been applied. The TSIS solar continuum is used.

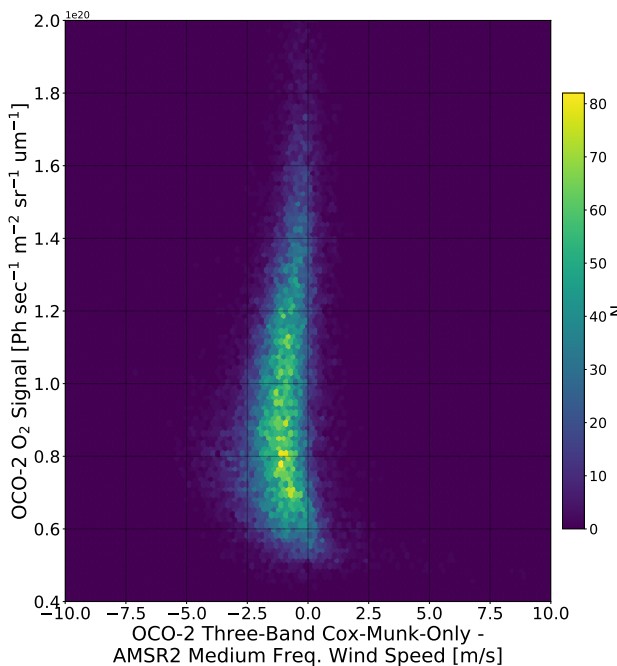

**Figure 5.** Heatmap of OCO-2 $O_2$ A-band signal compared to wind speed difference (OCO-2 three-band Cox-Munk-only - AMSR2 medium freq.)





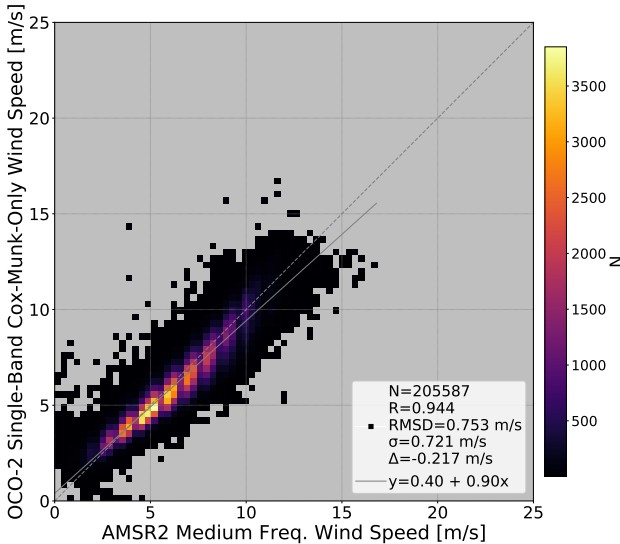

**Figure 6.** Heatmap of OCO-2 single-band Cox-Munk-only wind speed compared to AMSR2 medium frequency wind speed. The B9 lite file quality flag, custom filtering, and height scaling have been applied. The TSIS solar continuum is used.

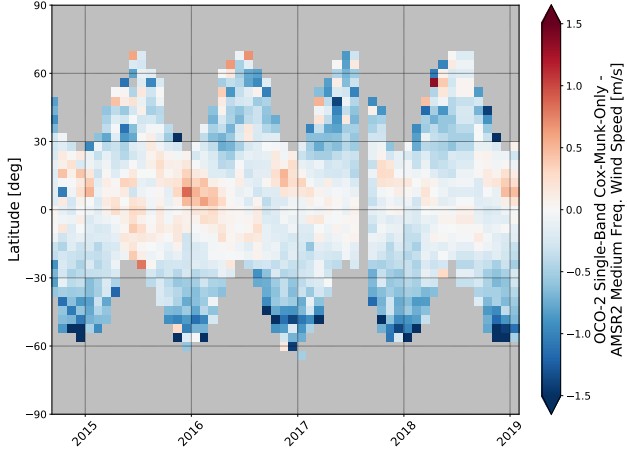

**Figure 7.** Heatmap of OCO-2 single-band Cox-Munk-only wind speed compared to AMSR2 medium frequency wind speed as a function of time and latitude. The B9 lite file quality flag, custom filtering, and height scaling have been applied. The TSIS solar continuum is used.