# Peer review of "Retrieved wind speed from the Orbiting Carbon Observatory-2"

_Atmospheric Measurement Techniques, 2020_

## Referee Comment (RC1) · Anonymous Referee #1 · 21 Jun 2020

This paper presents several different methods for retrieving sea surface wind speed from the Orbiting Carbon Observatory-2 (OCO-2), and then assesses the retrieved wind quality using the AMSR2 radiometer wind speed as reference. It shows a signle-band Cox-Munk-only model produces the best results as compared to AMSR2. Although there are many restrictions for the wind retrieval with infrared sensors, potential benefits of its high spatial resolution are drawing interest in ocean remote sensing applications. The manuscript is well-written and easy to follow.

My main concerns are as follows:

1. The spatial resolution of OCO-2 winds is not well addressed in the manuscript. Since OCO-2 and AMSR2 are actually resolving different size of surface wind characteristics, it is very relevant to discuss their spatial resolution and the associated inherent wind

variability.

2. The number of samples in Figs. 1, 2, 4, and 6 are different, making the conclusions of the comparison be less convincible. Do you compare the winds over the same area and the same period? I understand that may be caused by different quality control methods, but the authors had better present a more fair comparison, or describe it clearly in the text.

3. Previous studies (e.g., Wang and Zhang, 2010) show that Sun glint models with wind direction dependence has better performance than those without wind direction dependence in terms of the correlation coefficient between model and satellite measurements. Do you think the wind direction could be also an important factor relating to the latitude-dependent bias in Fig. 7? The isotropic CM model may work well in case the wind variability within the sensor footprint is large, such that the overall slope statistics become isotropic, right? And vice versa?

4. Many details about the methods are missed as the authors assume that the readers can understand all the details involved in the retrieval scheme. I think it's necessary to present more details in Section 3.

5. It is well-know that the sea surface slopes depend on the local wind, fetch, and incoming swell as well. I think it is necessary to pay attention on the segregation of wind sea and swell in the verification, before concluding an algorithm as scientifically credible.

Minor Comments:

6. lines 21-24: the overview of sea surface wind measurements from active radars and passive radiometers is insufficient. Some other scatterometers, altimeters and SARs are not mentioned in the text.

7. Fig. 1 and among others, the acronyms (R, RMSD, $\Delta,\sigma$) in the legend are not explained in the text or the caption.

---

## Referee Comment (RC2) · François-Marie Bréon (Referee) · 3 Jul 2020

The main purpose of the OCO-2 instrument is to retrieve the column integrated CO2 concentration (XCO2). Over water surfaces, the measurement are acquired in the glint mode where the instrument is pointed to a direction that is close to the glint geometry. The measurements are then sensitive to the sea surface reflectance which itself is sensitive to the wind speed. The measurement can then be used for an estimate of that parameter. Yet, the algorithm also includes a semi-empirical Lambertian component to the surface reflectance that is designed to represent calibration errors. This feature of the retrieval algorithm can be turned on or off. This paper analyse the precision and accuracy of the wind speed retrieval in comparison to the estimate from another instrument -assumed to be more precise- that flies over the same orbit as part of the A-

Train. It is shown that the wind speed error is rather large when using the operational algorithm with the Lambertian albedo feature, but that it gets much better when the feature is turned off (ie the surface reflectance is based on the glint reflectance only).

The wind speed retrieval uses the Cox-Munk model for the see surface slope distribution as a function of wind speed. It is said (correctly) that the model uses a Gram Charlier expansion (page 4, line 11). Yet, the equation line 14 strongly indicates that the author use the simplified version of the Cox and Munk model that is NOT based on a Gram-Charlier expansion and that do not depend on the wind direction. This is a strong assumption that must be discussed. Indeed, the wind speed direction has a very significant impact on the glint reflectance, in particular when the observation geometry is away from the glint. This may explain some of the features that are commented by the authors (but without referring to the wind direction)

In addition, it is stated (page 3, line 12) that the accuracy of the AMSR product is 1-1.5 m/s. Yet, the comparison of the best OCO-2 wind speed product against AMSR leads to a RMSD of 0.75, which is significantly lower than the stated accuracy. This indicates that the errors in the AMSR product and the OCO-2 product are significantly correlated, so that AMSR product cannot be used as an independent validation dataset. At the very least, this should be discussed.

One result of the paper is that the operational OCO-2 retrieval leads to wind speed estimated that are rather poor. This provides strong evidence that the Lambertian reflectance correction as a negative impact on some features of the retrieval. This is, I think, a result of importance that could be included in the abstract.

Minor issue : I suggest that the heat maps of Figure 3 and 5 use the same color table as those of the others, with grey color for values with no count.

---

## Author Comment (AC1) · 31 Aug 2020

See attachment.

Please also note the supplement to this comment:
https://amt.copernicus.org/preprints/amt-2020-180/amt-2020-180-AC1-supplement.pdf
* * *

---

## Author Comment (AC2)

The authors would like to thank François-Marie Bréon for their comments on our manuscript entitled, "Retrieved wind speed from the Orbiting Carbon Observatory-2." Below, we have addressed their comments and made the necessary changes in the manuscript.

**"The wind speed retrieval uses the Cox-Munk model for the see surface slope distribution as a function of wind speed. It is said (correctly) that the model uses a Gram Charlier expansion (page 4, line 11). Yet, the equation line 14 strongly indicates that the author use the simplified version of the Cox and Munk model that is NOT based on a Gram-Charlier expansion and that do not depend on the wind direction. This is a strong assumption that must be discussed. Indeed, the wind speed direction has a very significant impact on the glint reflectance, in particular when the observation geometry is away from the glint. This may explain some of the features that are commented by the authors (but without referring to the wind direction)"**

We have clarified the text in Section 3.1:

"They also found that the mean square slope parameter, which describes the surface roughness in their photographs, could be related to wind speed to a first order approximation using a simplified isotropic (independent of wind direction) function…"

We also examined the difference between the sensor azimuth angle and the wind direction (Fig. 1, below) and found no obvious correlation with the spatial errors (Fig. 2, below).

[Figure]

cos(2*(OCO-2 Met. Wind Direction - OCO-2 Sensor Azimuth Angle)), N=205587

*Figure 1. Values of 1 indicate the sensor and wind direction are parallel, while values of -1 indicate that they are perpendicular.*

[Figure]

OCO-2 Single-Band Cox-Munk-Only -
AMSR2 Medium Freq. Wind Speed [m/s], N=205587

*Figure 2. Difference between OCO-2 wind speed and AMSR2 wind speed.*

We have added the following statement to Section 5:

"Finally, the isotropic simplification of Cox-Munk used in our retrieval means that wind direction is not taken into account and thus the estimated wind speed could vary slightly depending on if the sensor is viewing up/downwind or crosswind. However, we analyzed the spatial patterns of the difference between the sensor azimuth angle and the meteorological wind direction (not used in the retrieval) and found no obvious correlation with the wind speed differences."

**"it is stated (page 3, line 12) that the accuracy of the AMSR product is 1-1.5 m/s. Yet, the comparison of the best OCO-2 wind speed product against AMSR leads to a RMSD of 0.75, which is significantly lower than the stated accuracy. This indicates that the errors in the AMSR product and the OCO-2 product are significantly correlated, so that AMSR product cannot be used as an independent validation dataset. At the very least, this should be discussed."**

Agreed, and we have added the following statement to Discussion and Conclusions:

"These errors are less than the estimated errors of AMSR2 itself (1-1.5 m/s), which may be partly because both sensors use similar assumptions about sea surface slope distributions and the relationship between these distributions, surface wind speed, and wind stress. Additionally, AMSR2 errors have typically been estimated by comparing to buoys, which has its own set of challenges including spatial-temporal matching errors, buoy height adjustment assumptions, and buoy measurement errors."

**"One result of the paper is that the operational OCO-2 retrieval leads to wind speed estimated that are rather poor. This provides strong evidence that the Lambertian reflectance correction as a negative impact on some features of the retrieval. This is, I think, a result of importance that could be included in the abstract."**

The Lambertian component of the retrieval has a clear positive impact on the $XCO_2$, which is the primary product from OCO-2. There are a number of retrieval setups that could potentially result in both accurate $XCO_2$ and wind speed, such as solving for wind speed in all three bands, but implementing and evaluating them was beyond the scope of this paper.

**"Minor issue : I suggest that the heat maps of Figure 3 and 5 use the same color table as those of the others, with grey color for values with no count."**

Figures 3 and 5 have been updated.

---

## Author Response (AR1)

The authors would like to thank Anonymous Referee #1 for their comments on our manuscript entitled, "Retrieved wind speed from the Orbiting Carbon Observatory-2." Below, we have addressed their comments and made the necessary changes in the manuscript.

**"The spatial resolution of OCO-2 winds is not well addressed in the manuscript. Since OCO-2 and AMSR2 are actually resolving different size of surface wind characteristics, it is very relevant to discuss their spatial resolution and the associated inherent wind variability."**

The following has been added to Section 5 (Discussion and Conclusions):

"As discussed in Sec.~\ref{sec:Data}, the footprint size of OCO-2 (1.25 km by 2 km at nadir) is much smaller than that of AMSR2 (0.25 degree latitude-longitude grid). Inherently, this means there will be more variability in the OCO-2 wind speeds. In order to quantify the impact, we calculated the overpass mean wind speed for each AMSR2 measurement. That is, the average wind speed of all OCO-2 footprints within a given AMSR2 grid cell. The difference statistics of this smoothed value compared to AMSR2 were similar to those of the unaveraged values, suggesting that the difference in spatial resolution of the two sensors does not significantly impact the overall results of this study."

"The number of samples in Figs. 1, 2, 4, and 6 are different, making the conclusions of the comparison be less convincible. Do you compare the winds over the same area and the same period? I understand that may be caused by different quality control methods, but the authors had better present a more fair comparison, or describe it clearly in the text."

As noted in Section 4.2, we down-sampled the much larger B9 dataset to check that the statistics were similar. We have modified it to read:

"The TSIS solar continuum test and the following retrieval modification tests were run on a relatively small dataset, but the statistics are similar when comparing the difference between OCO-2 B9 and AMSR2 on a comparably sampled dataset. This smaller dataset was specifically designed to cover the same temporal and spatial range as the full B9 dataset."

"Previous studies (e.g., Wang and Zhang, 2010) show that Sun glint models with wind direction dependence has better performance than those without wind direction dependence in terms of the correlation coefficient between model and satellite measurements. Do you think the wind direction could be also an important factor relating to the latitude-dependent bias in Fig. 7? The isotropic CM model may work well in case the wind variability within the

**sensor footprint is large, such that the overall slope statistics become isotropic, right? And vice versa?"**

We examined the difference between the sensor azimuth angle and the wind direction (Fig. 1, below) and found no obvious correlation with the spatial errors (Fig. 2, below).

cos(2\*(OCO-2 Met. Wind Direction - OCO-2 Sensor Azimuth Angle)), N=205587 Figure 1. Values of 1 indicate the sensor and wind direction are parallel, while values of -1 indicate that they are perpendicular.